# Local Control Networks (LCNs): Optimizing Flexibility in Neural Network Data Pattern Capture

## Abstract

The widespread use of Multi-layer perceptrons (MLPs) often relies on a fixed activation function (e.g., ReLU, Sigmoid, Tanh) for all nodes within the hidden layers. While effective in many scenarios, this uniformity may limit the network's ability to capture complex data patterns. We argue that employing the same activation function at every node is suboptimal and propose leveraging different activation functions at each node to increase flexibility and adaptability. To achieve this, we introduce Local Control Networks (LCNs), which leverage B-spline functions to enable distinct activation curves at each node. Our mathematical analysis demonstrates the properties and benefits of LCNs over conventional MLPs. In addition, we demonstrate that more complex architectures, such as Kolmogorov–Arnold Networks (KANs), are unnecessary in certain scenarios, and LCNs can be a more efficient alternative. Empirical experiments on various benchmarks and datasets validate our theoretical findings. In computer vision tasks, LCNs achieve marginal improvements over MLPs and outperform KANs by approximately 5%, while also being more computationally efficient than KANs. In basic machine learning tasks, LCNs show a 1% improvement over MLPs and a 0.6% improvement over KANs. For symbolic formula representation tasks, LCNs perform on par with KANs, with both architectures outperforming MLPs. Our findings suggest that diverse activations at the node level can lead to improved performance and efficiency.

## 1 Introduction

Multi-layer perceptrons (MLPs) have achieved remarkable success across various domains, from computer vision to natural language processing. Traditionally, networks have relied on a fixed activation function throughout their architecture, with popular choices including ReLU, Sigmoid, and Tanh (Dubey et al., 2022). The ReLU activation function has gained widespread popularity due to its simplicity and effectiveness, especially in addressing the vanishing gradient problem (Petersen & Voigtlaender, 2018). However, its non-smooth nature can lead to issues such as "dead neurons," where neurons become inactive and no longer contribute to the learning process. On the other hand, smooth activation functions such as sigmoid and tanh, while easier to optimize due to their differentiability, are prone to the vanishing gradient problem, especially in deeper networks (Petersen & Voigtlaender, 2018). In addition to the specific weaknesses of each activation function, applying the same activation function across all nodes can limit flexibility and expressive power. This uniformity means that all neurons process information in the same way, potentially restricting the network's ability to capture diverse and complex patterns in the data. Furthermore, using fixed activation functions causes every weight to be updated whenever a new data point is introduced, affecting the weights learned from previous training data.

The hypothesis of using different activation functions at distinct nodes has inspired the development of Local Control Networks (LCNs), as proposed in this paper. Recently, KANs have been introduced as a method to address the limitations of fixed activation functions by proposing learnable activation functions and putting it in the edge (Liu et al., 2024). However, while KANs offer increased flexibility, they still face certain limitations in their application. In contrast, LCNs aim to further enhance flexibility and representational capacity by allowing multiple activation functions to coexist within a single network, enabling the model to adapt more effectively to varying data characteristics

(Hagg et al., 2017). This approach is based on the idea that different neurons or layers may benefit from specific activation functions, depending on the features they are processing. To achieve this diversity in activation functions, we chose the B-Spline as the activation function for LCNs. In our network design, the parameters of the B-Spline activation functions are learnable, enabling LCNs to adjust activation functions at individual nodes dynamically. Additionally, learnable B-spline functions support localized adjustments. This means the B-spline curves can be updated locally to adapt to new data without altering the entire function trained on previous data, thereby improving stability and speeding up convergence. We also hypothesize that this use of diverse and locally adaptive activation functions will not only enhance model performance but also improve interpretability. The shape of the B-Spline function at each neuron reflects the data patterns the neuron captures, offering insights into the specific contributions of individual neurons, enhancing the interpretability of the network (Fakhoury et al., 2022). For instance, in image classification tasks, individual neurons can specialize in detecting features like edges, textures, or shapes. The learned B-spline function at each neuron directly reflects these localized patterns, making it easier to trace which neuron is responsible for capturing specific features. This capability enhances transparency, as we can analyze individual neuron responses to understand what each is detecting.

An overview of the limitations in conventional MLPs and the corresponding solutions of LCNs is shown in Figure 1.

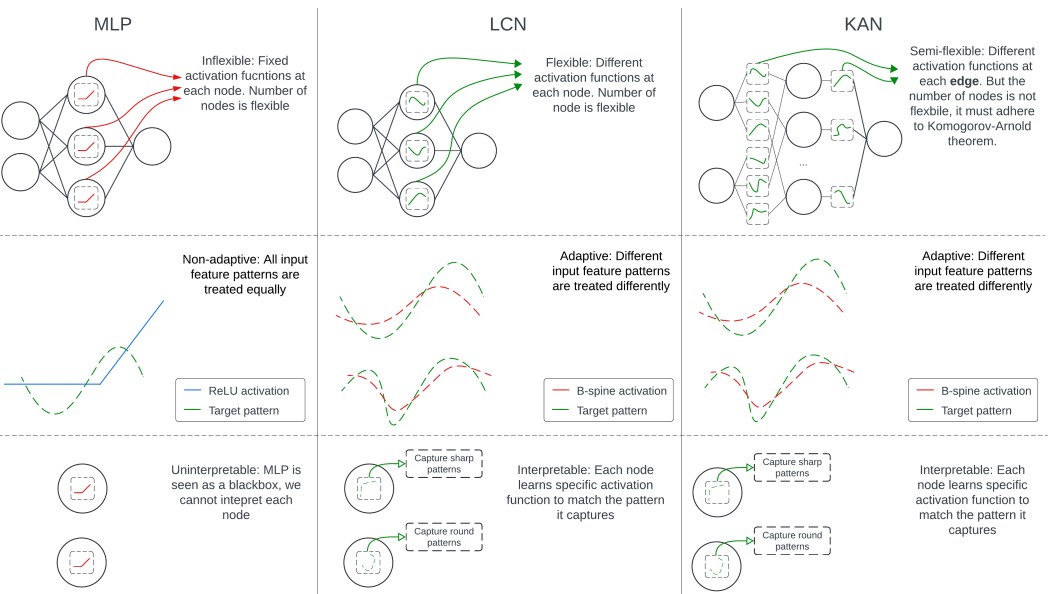

Figure 1: Limitations of conventional DNNs and the corresponding improvements from LCNs.

To validate our hypothesis, we conducted a theoretical analysis and empirical experiments on a Local Control Networks (LCNs) and compared it with other alternatives, such as conventional Multi-Layer Perceptrons (MLPs) and Kolmogorov–Arnold Networks (KANs). The analysis and experimental results reveal that utilizing different activations and the local support property within LCNs enhances not only the network's ability to capture complex data patterns but also its stability, convergence speed, and interpretability. The remainder of this paper is organized as follows: Section 2 provides an overview of the related work. Section 3 introduces the mathematical analysis and design of LCNs. Section 4 highlights the effectiveness of LCNs over conventional MLPs and KANs. Section 5 demonstrates the experimental setup, results and discussion. We end the paper with conclusion and insights in Section 6.

## 2 RELATED WORK

The development and optimization of activation functions have been extensively studied in the field of neural networks. Conventional activation functions such as ReLU, Sigmoid, and Tanh have been pivotal in the success of deep learning models. However, each of these functions comes with inherent

limitations. ReLU, for example, mitigates the vanishing gradient problem but suffers from dead neurons and lacks the smoothness needed for capturing subtle patterns (Dubey et al., 2022; Petersen & Voigtlaender, 2018; Laurent & Brecht, 2018). Smoother functions like Sigmoid and Tanh face slow convergence and vanishing gradients, especially in deep architectures.

To address the limitations of those activation functions, various researchers have proposed adaptive or learnable activation functions. For instance, Swish (Ramachandran et al., 2017) and Mish (Misra, 2019) are non-monotonic and smooth activations that have demonstrated improvements in a variety of tasks. These functions improve gradient flow and generalization, highlighting the need for more adaptable activation mechanisms. Furthermore, the concept of learning activation functions during training has been explored (Arora et al., 2018), allowing networks to adapt activation behaviors to specific datasets.

Using varied activation functions within the same network leverages their combined strengths. Hagg et al. (2017) extended the NEAT algorithm to evolve networks with heterogeneous activation functions, demonstrating that such networks can outperform homogeneous ones while remaining smaller and more efficient. Dushkoff & Ptucha (2016) introduced networks where each neuron can select its activation function from a predefined set, learning both weights and activations during training.

The use of spline-based activation functions introduces a new level of flexibility to neural networks. B-spline activations stand out for their ability to smoothly approximate complex data distributions. B-splines, known for their flexibility and precision in numerical analysis and computer graphics, are ideal for neural networks that need to accurately model intricate patterns. Recent research highlights their effectiveness in various machine learning tasks. For example, Bohra et al. (2020) developed a framework for optimizing spline activation functions during training, improving both accuracy and interpretability. Similarly, the ExSpliNet model Fakhoury et al. (2022) combines Kolmogorov networks, probabilistic trees, and multivariate B-splines for improved interpretability and performance. This research demonstrates the potential of spline-based activations to enhance both performance and clarity in deep learning models.

Despite their capacity to overfit, deep networks often generalize well, challenging traditional views like the bias-variance trade-off (Belkin et al., 2019). Studies show networks can fit random labels yet generalize on real data (Zhang et al., 2017). The Lottery Ticket Hypothesis (Frankle & Carbin, 2019) and research on intrinsic dimensionality (Li et al., 2018) further explain how over-parameterized models generalize effectively, prompting a reevaluation of complexity in neural networks.

The Kolmogorov–Arnold representation theorem underpins the expressive power of neural networks and their approximation abilities. Liu et al. (2024) expanded on this theorem by introducing Kolmogorov–Arnold Networks (KANs), which leverage spline-based activation functions to achieve higher levels of accuracy and interpretability compared to traditional models. Kolmogorov–Arnold theorem is particularly notable for their ability to overcome the curse of dimensionality by decomposing complex high-dimensional functions into compositions of simpler one-dimensional functions (Schmidt-Hieber, 2021). This approach allows KANs to build more compact and interpretable models, making them a promising direction for future research in deep learning.

Building upon these works, our proposed LCNs differ by enabling each neuron to have a unique, learnable B-spline activation function, offering a higher degree of flexibility and local adaptability compared to networks with fixed or globally adaptive activation functions. Unlike previous methods that may require complex architectures or evolutionary algorithms, LCNs maintain a standard network structure with enhanced activation capabilities, making them practical for a wide range of applications.

## 3  PROPOSED METHODOLOGY

In this paper, we introduce the Local Control Networks (LCNs), a neural network architecture that leverages B-spline functions to enable different activation functions at each node. We provide a comprehensive mathematical analysis demonstrating the advantages of LCNs over conventional MLPs and more complex architectures Kolmogorov–Arnold Networks (KANs). Our approach simplifies network design while enhancing expressiveness, data capture flexibility, computational efficiency, and generalization capabilities.

## 3.1 BACKGROUNDS

MLPs are a class of feed forward artificial neural networks that have been widely used for function approximation and pattern recognition tasks. In traditional MLPs, each neuron in a layer applies a fixed activation function—commonly the ReLU or Sigmoid function—to its weighted input sum. While these fixed activation functions simplify the network design and training process, they introduce limitations in capturing complex and nuanced patterns in data. Specifically, fixed activations can lead to issues such as vanishing gradients, lack of smoothness, and inability to model localized features effectively (Dubey et al., 2022). Fixed activation functions impose a uniform response across all neurons, which can hinder the network's flexibility and expressiveness. This motivates the exploration of more adaptable activation mechanisms that can enhance the network's capacity to model complex functions.

### 3.1.1 ACTIVATION FUNCTION DIVERSITY

To address the limitations of fixed activation functions, we propose using diverse activation functions for each neuron, enabling the network to flexibly model a wider range of data patterns and capture both global trends and local variations.

B-spline functions are well-suited for this purpose due to their smoothness, continuity, and local support properties, which ensure that a neuron's activation is influenced only by localized input regions. These properties enhance the network's ability to model complex, localized features effectively. B-splines, widely used in numerical analysis and approximation theory, offer smooth and localized approximations, making them ideal for developing adaptive activation functions that improve performance and pattern recognition (Bohra et al., 2020; Lyche et al., 2018; De Boor, 1978).

Let $\mathcal{E} = \{\mathcal{E}_1, \mathcal{E}_2, \ldots, \mathcal{E}_r\}$ be the non-decreasing knot sequence where the knots satisfy $\mathcal{E}_1 \leq \mathcal{E}_2 \leq \cdots \leq \mathcal{E}_r$. The number of B-splines of degree $p$ is defined as $N = r - p - 1$, where $p \geq 0$ is the degree of the B-spline and $r$ is the number of knots.

**Base Case**: Degree $p = 0$ The B-spline of degree $p = 0$ is given by:

$$B_{\mathcal{E},0,n}(x) = \begin{cases} 1 & \text{if } x \in [\mathcal{E}_n, \mathcal{E}_{n+1}), \\ 0 & \text{otherwise.} \end{cases} \tag{1}$$

**Recursive Definition**: Degree $p \geq 1$ For $p \geq 1$, the B-spline basis function is defined recursively as:

$$B_{\mathcal{E},p,n}(x) = \frac{x - \mathcal{E}_n}{\mathcal{E}_{n+p} - \mathcal{E}_n} B_{\mathcal{E},p-1,n}(x) + \frac{\mathcal{E}_{n+p+1} - x}{\mathcal{E}_{n+p+1} - \mathcal{E}_{n+1}} B_{\mathcal{E},p-1,n+1}(x). \tag{2}$$

## 3.2 LOCAL CONTROL NETWORKS ARCHITECTURE

In Local Control Networks (LCN), the network employs B-spline-based activation functions across multiple layers, which enable it to capture complex data structures more effectively compared to traditional activations like ReLU. B-splines provide smooth, continuous mappings, offering a more accurate approximation of the underlying data manifold and avoiding sharp transitions inherent in piecewise linear functions like ReLU. The Figure 2 visually represents how information flows through each layer, with each neuron utilizing distinct B-spline activation functions.**

### 3.2.1 INPUT LAYER

The input to the network is denoted as $x = [x_1, x_2, \ldots, x_D] \in [0, 1]^D$, where $D$ represents the dimensional of the input space. Each input feature $x_d$ belongs to a unit interval $[0, 1]$.

### 3.2.2 HIDDEN LAYERS

The network consists of $L$ hidden layers. Each hidden layer $l \in \{1, \ldots, L\}$ has $M_l$ neurons, and the activations in each layer are computed using B-spline functions as activation functions. For neuron $i$ in layer $l$, the activation is given by:

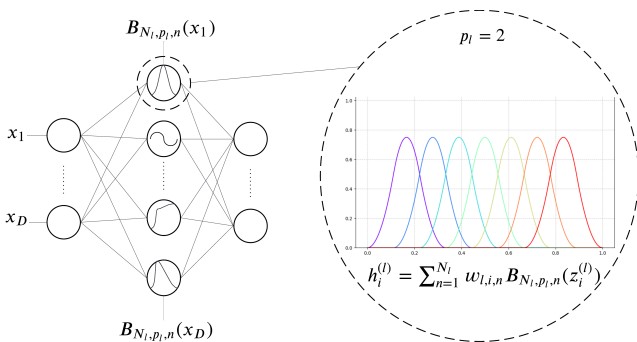

Figure 2: Local Control Network Architecture with B-spline Activation Functions.

$$h_i^{(l)} = \sum_{n=1}^{N_l} w_{l,i,n} B_{N_l,p_l,n}(z_i^{(l)}) \tag{3}$$

where $B_{N_l,p_l,n}$ is the $n^{th}$ B-spline basis function of degree $p_l$ applied to $z_i^{(l)}$. $w_{l,i,n}$ are the learned weights associated with the $n^{th}$ B-spline basis function for neuron $i$ in layer $l$. $z_i^{(l)}$ is the linear transformation for the $i^{th}$ neuron in layer $l$. $N_l$ is the number of B-spline basis functions used for each neuron in layer $l$.

B-spline functions offer smooth and continuous transitions, ensuring stability in learning and capturing complex patterns in data. The local support property of B-splines enables selective activation, which contributes to efficient learning and avoids global interference from irrelevant neurons.

### 3.2.3 OUTPUT LAYER

The output layer has $O$ neurons (corresponding to the number of outputs), and the final output $\hat{y}$ is computed as:

$$\hat{y} = W^{(L+1)} h^{(L)} + b^{(L+1)} \tag{4}$$

where $W^{(L+1)} \in \mathbb{R}^{O \times N_L}$ is the weight matrix for the output layer. $b^{(L+1)} \in \mathbb{R}^O$ is the bias vector for the output layer. $h^{(L)}$ is the activation vector from the last hidden layer.

### 3.3 GRADIENT COMPUTATION

Gradient computation plays a crucial role in training LCN, as it ensures efficient weight updates during backpropagation. The use of B-splines introduces some unique properties in gradient computation that we now explore.

### 3.3.1 LOSS FUNCTION

In neural networks, we generally minimize a loss function that measures the difference between predicted and true values. For our formulation, let $L(y, \hat{y})$ represent a general convex loss function, where $y$ is the true label and $\hat{y}$ is the predicted output. To simplify and demonstrate our methodology, we employ the commonly used Mean Squared Error (MSE) as the loss function.

### 3.3.2 GRADIENT WITH RESPECT TO INPUT VARIABLES

The gradient of the loss function $L$ with respect to the input variables provides insights into how input changes affect the output. Using the chain rule, the gradient with respect to $x_d$ is computed as:

$$\frac{\partial L}{\partial x_d} = \sum_{l=1}^{L} \sum_{i=1}^{M_l} \frac{\partial L}{\partial h_i^{(l)}} \cdot \frac{\partial h_i^{(l)}}{\partial z_i^{(l)}} \cdot \frac{\partial z_i^{(l)}}{\partial x_d} \tag{5}$$

which extends to:

$$\frac{\partial L}{\partial x_d} = \sum_{l=1}^{L} \sum_{i=1}^{M_l} \frac{2}{m} (\hat{y}_i - y_i) \sum_{n=1}^{N_l} w_{l,i,n} \frac{\partial B_{N_l,p_l,n}(z_i^{(l)})}{\partial z_i^{(l)}} W_{id}^{(1)} \tag{6}$$

**Key Observations:**

**a) Local Support of B-splines**  The derivative $\frac{\partial B_{N_l,p_l,n}(z_i^{(l)})}{\partial z_i^{(l)}}$ is non-zero only within a specific interval due to the local support property of B-splines. This means that only a subset of neurons contributes to the gradient at any given input.

**b) Dependence on Input Weights**  The gradient depends directly on the weights from the input layer to the neurons activated by the input.

**c) Robustness to Input Changes**  Since the gradient depends only on the local region where the B-spline is active, small changes or scaling in the input $x_d$ outside this region have minimal impact on the loss. This leads to a network that is more robust to input perturbations.

**d) Effective Dropout Mechanism**  The localized gradient naturally ignores irrelevant inputs, similar to the effect of dropout. Neurons are selectively activated based on the input, enhancing the network's ability to generalize and reducing overfitting.

### 3.3.3 GRADIENT WITH RESPECT TO WEIGHT PARAMETERS

The gradient of the loss $L$ with respect to the weight parameters $W_{ij}^{(l)}$ is computed as:

$$\frac{\partial L}{\partial W_{ij}^{(l)}} = \frac{\partial L}{\partial h_i^{(l)}} \cdot \frac{\partial h_i^{(l)}}{\partial z_i^{(l)}} \cdot \frac{\partial z_i^{(l)}}{\partial W_{ij}^{(l)}} \tag{7}$$

The final gradient of the loss with respect to the weight parameters $W_{ij}^{(l)}$ is showed:

$$\frac{\partial L}{\partial W_{ij}^{(l)}} = \frac{2}{m} (\hat{y}_i - y_i) \sum_{n=1}^{N_l} w_{l,i,n} \frac{\partial B_{N_l,p_l,n}(z_i^{(l)})}{\partial z_i^{(l)}} \cdot h_j^{(l-1)} \tag{8}$$

**Key Observations:**

**a) Selective Activation**  Due to the local support, only certain neurons with activations within the B-splines support contribute significantly.

**b) Localized Weight Updates**  The gradient is substantial only for weights connected to neurons that are both active and within the B-splines support region. Only relevant weights are updated during backpropagation, reducing computational overhead and leading to faster convergence.

**c) Sparse Updates for Efficiency**  Only a subset of weights receive significant updates during backpropagation. This sparsity reduces computational overhead and leads to faster convergence during training.

**d) Smooth Optimization Landscape**  The smoothness of B-spline functions results in well-behaved gradients, avoiding issues like vanishing or exploding gradients common in networks with non-smooth activation functions.

## 4 COMPARATIVE INSIGHTS

The Local Control Networks (LCNs) offers a unique balance between simplicity and expressiveness in comparison to both traditional MLPs with fixed activation and more complex architecture such as KANs. This section outlines the core insights derived from LCN's design, highlighting its advantages in performance, generalization, and computational efficiency by comparing it to MLPs and KANs.

### 4.1 LOCALIZED ACTIVATION AND SMOOTH GRADIENTS

One of the key features of LCN is the local support property of B-spline activation functions. This provides several advantages over traditional activations such as ReLU:

- ReLU, being piecewise linear, introduces sharp transitions at zero, which can destabilize training and result in gradients influenced by broad input ranges, reducing efficiency.

- B-splines, in contrast, ensure smooth transitions and localized support, confining input influence to specific regions. This leads to sparse gradient updates, reduced computational complexity, and improved stability during training, akin to an effective dropout mechanism.

Thus, LCN can handle subtle variations in data more efficiently than ReLU, providing stability and robustness against noise and small perturbations in the input data.

### 4.2 EFFICIENCY AND COMPUTATIONAL LOAD

Kolmogorov–Arnold Networks (KAN) offer strong theoretical guarantees for function approximation but suffer from high computational complexity due to their reliance on combining multiple univariate functions, leading to overparameterization and inefficiency in large-scale applications.

In contrast, LCN retains the simplicity of standard neural architectures while leveraging B-splines for flexible, accurate approximations without the need for complex transformations. Localized weight updates in LCN enhance computational efficiency by reducing simultaneous parameter updates, resulting in faster convergence, sparser computations, and lower training time and memory usage, all while effectively modeling complex patterns.

### 4.3 FUNCTION APPROXIMATION AND GENERALIZATION

LCN strikes a balance between expressiveness and generalization. The use of B-spline activations allows LCN to approximate complex functions with smooth, localized responses, which reduces the risk of overfitting. The localized support of B-splines ensures that the network focuses only on relevant regions of the input space, effectively acting as a natural regularize. This leads to a model that can generalize well across different datasets, even in cases where high-dimensional noise might cause KAN to falter. LCN's localized gradient updates also contribute to this generalization by ensuring that irrelevant inputs are ignored, much like a built-in regularization mechanism.

### 4.4 SIMPLIFIED ARCHITECTURE FOR PRACTICAL USE

From a practical perspective, LCN offers a significant advantage over KAN by maintaining a standard neural network architecture augmented with B-spline activations. This results in a simpler and more scalable design, which can be easily implemented in existing neural network frameworks without requiring the complex setup associated with KAN's univariate function combinations.

ReLU, though computationally efficient, is limited in its ability to model more nuanced data structures due to its lack of smoothness and piecewise linearity. LCN, on the other hand, provides a smooth function approximation through B-splines, avoiding the sharp transitions characteristic of ReLU, and enabling the network to model more intricate data manifolds. B-splines allow the network to capture both small and large variations in the input space, leading to more robust learning and eliminating the problem of neuron saturation commonly seen with ReLU.

## 5 EXPERIMENT

We conducted experiments to validate our theoretical findings and evaluate the practical performance of LCN compared to MLPs with fixed activations and KANs. We tested the models on various benchmark datasets, focusing on accuracy, convergence speed, and computational efficiency.

### 5.1 EXPERIMENTAL SETUP

#### 5.1.1 DATASETS

We selected a diverse set of datasets to cover different types of tasks, following the standards set in (Yu et al., 2024):

- **Basic Machine Learning Tasks**: Bank Marketing, Bean Classification, Spam Detection, and Telescope datasets.
- **Computer Vision Tasks**: MNIST and Fashion-MNIST (FMNIST) for image classification.
- **Symbolic Representation Tasks**: Synthetic datasets for symbolic regression.

#### 5.1.2 MODEL CONFIGURATIONS

To ensure a fair comparison, we standardized the total number of parameters by adjusting the layers and neurons across all models. LCN and KAN underwent hyperparameter tuning via grid search for optimal performance.

### 5.2 RESULTS AND DISCUSSION

#### 5.2.1 BASIC MACHINE LEARNING TASKS

In this section, we evaluate the performance of MLP, KAN, and LCN models on four different basic machine learning tasks. Figure 3 shows that LCN consistently have higher accuracy than both MLP and KAN while maintaining efficient use of parameters.

#### 5.2.2 COMPUTER VISION TASKS

On the MNIST and FMNIST datasets, LCN achieved slight improvements over MLP and outperformed KAN by approximately 5%. LCN also demonstrated better computational efficiency, particularly in handling high-dimensional input spaces, while KAN struggled to scale effectively. Figure 4 presents a comparison of the accuracy trends on these datasets for MLP, KAN, and LCN models.

Figures 3 and 4 use a range of different model parameters to examine each model's performance scaling relative to its architecture. To approximate fairness, we carefully adjust width, depth, and key architectural components (e.g., B-spline grids and order parameters in KANs and LCNs). By varying parameters in a controlled manner, Figures 3 and 4 highlight how each model type leverages parameter increases, offering valuable insights into the scalability and efficiency of different architectures. The number of parameters for MLPs is capped because they achieve convergence at lower parameter counts than LCNs and KANs, beyond which their performance does not improve. Extending the parameter range for MLPs would add no additional insights since their accuracy plateaus.

#### 5.2.3 SYMBOLIC REPRESENTATION TASKS

We evaluate the performance of MLP, KAN, and LCN models on symbolic regression tasks, which involve approximating mathematical functions. Figure 5 compares their accuracy against the number of parameters.

KAN excels due to its univariate function decomposition but has higher computational costs and slower convergence. LCN matches or surpasses KAN's accuracy with fewer parameters and faster convergence, thanks to its localized B-spline activation functions. The B-spline's localized support enables efficient learning and scalability, making LCN a practical alternative for tasks requiring both accuracy and efficiency.

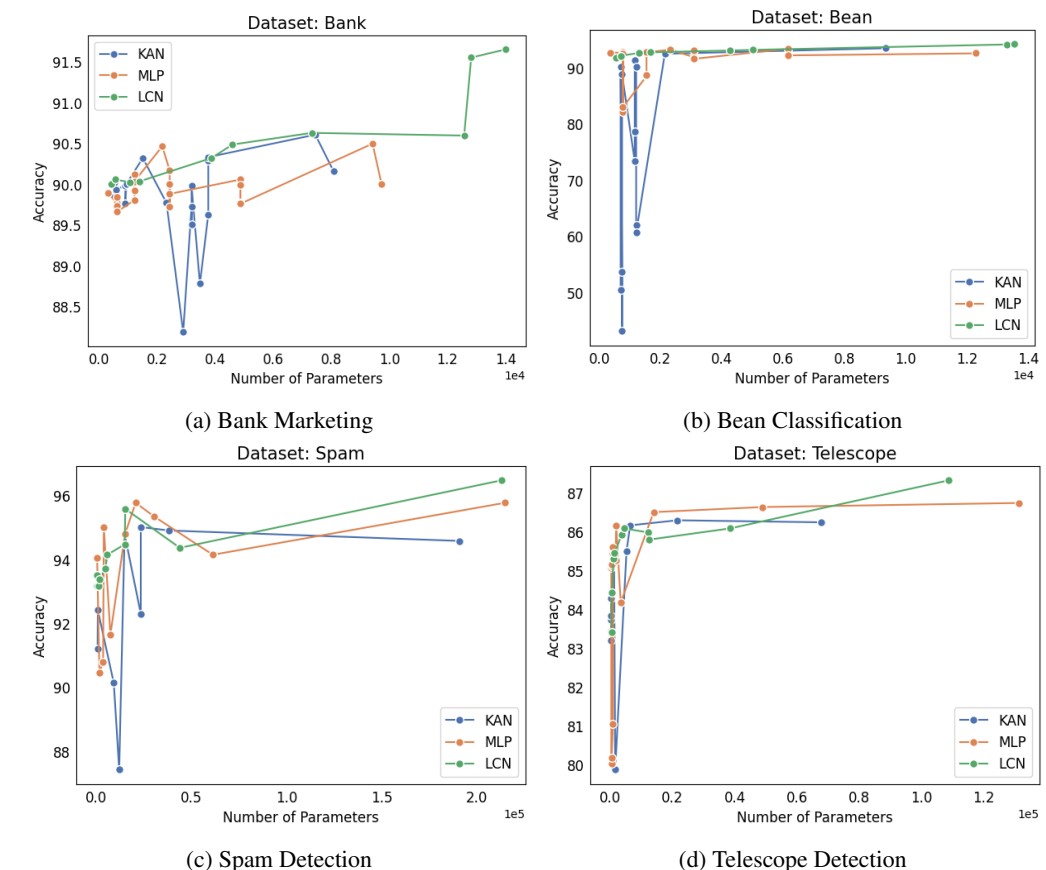

Figure 3: Comparison of accuracy over the number of parameters for MLP, KAN, and LCN models on four datasets.

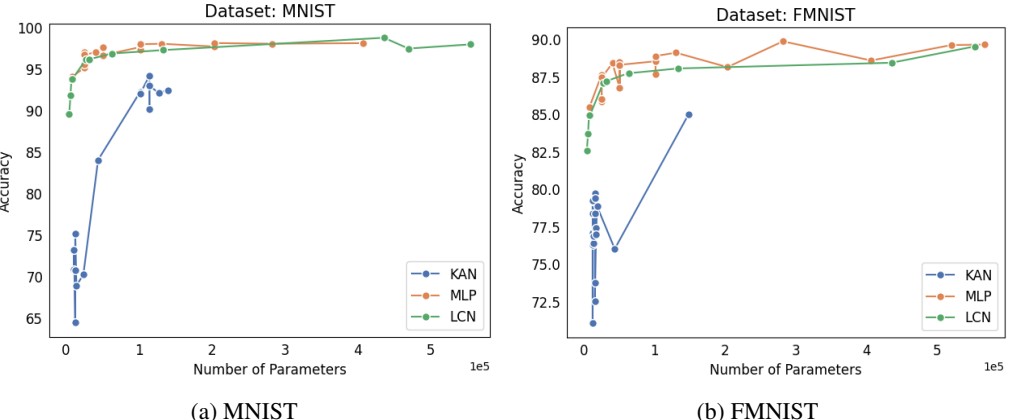

Figure 4: Comparison of accuracy over the number of parameters for MLP, KAN, and LCN models on MNIST and FMNIST datasets.

### 5.2.4 CONVERGENCE SPEED AND LEARNING CURVES

Despite starting with a lower initial accuracy, LCN exhibited faster learning, reaching higher accuracy within the first few epochs compared to MLP and KAN models. This rapid convergence demonstrates LCN's ability to efficiently capture data patterns using its localized activation func-

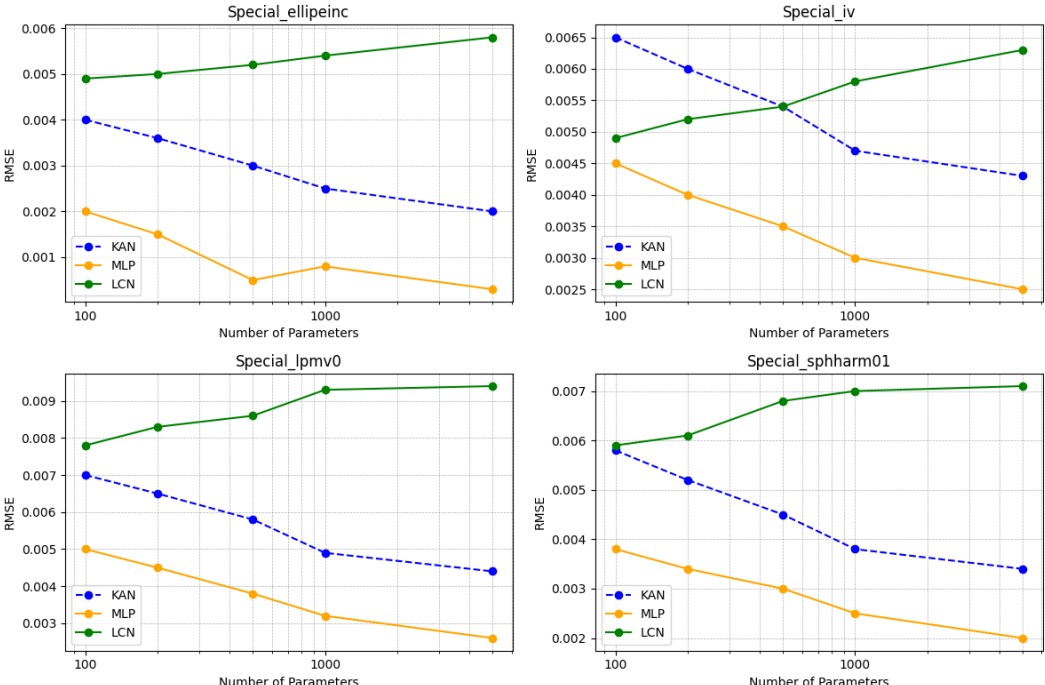

Figure 5: Comparison of accuracy over the number of parameters for MLP, KAN, and LCN models on symbolic functions.

tions. In contrast, KAN showed slower convergence and lower overall accuracy, likely due to its more complex and computationally demanding architecture.

### 5.2.5 DISCUSSION

The marginal improvements of LCNs over MLPs and KANs reflect the simplicity of the datasets used, which follow the standards in (Yu et al., 2024). On these basic tasks, MLPs already achieve strong performance, leaving limited room for improvement for LCNs and KANs. However, the flexibility and localized activations of LCNs are expected to show greater advantages on more complex datasets with higher-dimensional inputs or intricate patterns. In the future work, we will focus on such challenging tasks to better demonstrate the potential of LCNs.

## 6 CONCLUSION

The use of diverse activation functions in Local Control Networks (LCNs) represents a major innovation in neural network design. Unlike traditional models that use a uniform activation function, LCNs allow each neuron to dynamically select the most suitable activation function, enhancing adaptability and expressiveness. This flexibility enables LCNs to effectively capture both global and localized data patterns, improving accuracy and efficiency across various tasks.

LCNs also achieve faster learning and better generalization by tailoring activation functions to different input regions. This dynamic adjustment leads to quicker convergence and makes LCNs a computationally efficient alternative to complex models like KANs, without requiring their intricate architectures.

In summary, LCNs enhance performance and scalability by leveraging flexible activation functions, providing a simpler yet powerful solution for diverse machine learning tasks.

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

# A APPENDIX

## A.1 THE CONSCIOUSNESS PRIOR

Bengio (2017) introduced the concept of the Consciousness Prior, suggesting that deep learning models could benefit from mechanisms that mimic the conscious selection and broadcasting of information, akin to human cognitive processes. This idea proposes a hierarchical structure within neural networks, where specific elements are made globally available to influence perception and decision-making processes. While still a theoretical construct, the Consciousness Prior points to the potential for neural networks to incorporate more sophisticated, human-like decision-making processes, opening new avenues for research in AI.

## A.2 B-SPLINE PRELIMINARY

B-splines are piecewise polynomial functions widely used in numerical analysis, computer graphics, and approximation theory. Their key properties—smoothness, continuity, and local support—make them valuable for enhancing neural network architectures, particularly in developing complex activation functions. This can lead to improved performance and pattern recognition (Bohra et al., 2020). We define B-splines and outline their properties for uniform partitions, drawing from (Lyche et al., 2018) and (De Boor, 1978), to support the use in the LCN model.

### A.2.1 PROPERTIES OF B-SPLINES

B-splines possess several key properties:

- **Non-negativity and Partition of Unity:** $B_{i,p}(x) \geq 0$ for all $x$, and $\sum_i B_{i,p}(x) = 1$ for all $x$.
- **Local Support:** $B_{i,p}(x)$ is non-zero only on the interval $[\xi_i, \xi_{i+p+1})$, which leads to sparse representations and efficient computations.
- **Continuity:** B-splines of degree $p$ are $C^{p-1}$ continuous, providing smooth approximations.
- **Derivative:** For $p \geq 1$, the derivative of a B-spline basis function is given by:

$$\frac{d}{dx^+} B_{i,p}(x) = p \left( \frac{B_{i,p-1}(x)}{\xi_{i+p} - \xi_i} - \frac{B_{i+1,p-1}(x)}{\xi_{i+p+1} - \xi_{i+1}} \right). \tag{9}$$

### A.2.2 LINEAR COMBINATION OF B-SPLINES

A spline function $S(x)$ can be expressed as a linear combination of B-spline basis functions:

$$S(x) = \sum_i w_i B_{i,p}(x), \tag{10}$$

where $w_i \in \mathbb{R}$ are the coefficients (weights).

This formulation allows for the approximation of any continuous function on a closed interval by adjusting the weights $w_i$, the degree $p$, and the knot sequence $\{\xi_i\}$ (De Boor, 1978).

### A.2.3 LINEAR TRANSFORMATION:

In each layer $l$, the input from the previous layer $h^{(l-1)}$ is linearly transformed using a weight matrix $W^{(l)}$ and bias vector $b^{(l)}$. The linear transformation for the $i^{th}$ neuron in layer $l$ is given by:

$$z_i^{(l)} = \sum_{j=1}^{M_{l-1}} W_{ij}^{(l)} h_j^{(l-1)} + b_i^{(l)} \tag{11}$$

where:

- $W_{ij}^{(l)} \in \mathbb{R}^{M_l \times M_{l-1}}$ is the weight matrix connecting $j^{th}$ neuron in $(l-1)^{th}$ layer to $i^{th}$ neuron in $l^{th}$ layer,

- $b^{(l)} \in \mathbb{R}^{M_l}$ is the bias vector for the $l^{th}$ layer,

- $h_j^{(l-1)}$ is the activation of the $j^{th}$ neuron in the $(l-1)^{th}$ layer,

- $z_i^{(l)}$ is the pre-activation of the $j^{th}$ neuron in layer $l$.

- $M_l$ is the number of neurons in layer $l$.

### A.2.4 GRADIENT WITH RESPECT TO INPUT VARIABLES

The gradient of the loss function with respect to the input variables is crucial for understanding how changes in the input affect the output prediction. We start by applying the chain rule, which propagates gradients from the output layer back to the input layer.

The gradient of the loss $L$ with respect to the input $x_d$ can be computed using the chain rule as written as:

$$\frac{\partial L}{\partial x_d} = \sum_{l=1}^{L} \sum_{i=1}^{M_l} \frac{\partial L}{\partial h_i^{(l)}} \cdot \frac{\partial h_i^{(l)}}{\partial z_i^{(l)}} \cdot \frac{\partial z_i^{(l)}}{\partial x_d} \tag{12}$$

Breaking it down into three terms:

(a) **Gradient of Loss with Respect to Activation:**

$$\frac{\partial L}{\partial h_i^{(l)}} = \frac{2}{m}(\hat{y}_i - y_i) \tag{13}$$

This term reflects the gradient of the loss function with respect to the activation $h_i^{(l)}$, which represents the output of the activation function for neuron $i$ in layer $l$.

(b) **Gradient of Activation with Respect to Pre-activation:** Since we are using B-spline activations, the gradient of the activation $h_i^{(l)}$ with respect to the pre-activation $z_i^{(l)}$ is:

$$\frac{\partial h_i^{(l)}}{\partial z_i^{(l)}} = \sum_{n=1}^{N_l} w_{l,i,n} \frac{\partial B_{N_l,p_l,n}(z_i^{(l)})}{\partial z_i^{(l)}} \tag{14}$$

The derivative of the B-spline basis function is given by:

$$\frac{\partial B_{N_l,p_l,n}(z_i^{(l)})}{\partial z_i^{(l)}} = p_l \left( \frac{B_{N_l,p_l-1,n}(z_i^{(l)})}{\xi_{n+p_l} - \xi_n} - \frac{B_{N_l,p_l-1,n+1}(z_i^{(l)})}{\xi_{n+p_l+1} - \xi_{n+1}} \right) \tag{15}$$

This term captures how the activation function changes in response to the linear combination of inputs before passing through the activation function.

(c) **Gradient of Pre-activation with Respect to Input:** The gradient of the pre-activation $z_i^{(l)}$ with respect to the input variable $x_d$ for the first layer is:

$$\frac{\partial z_i^{(l)}}{\partial x_d} = W_{id}^{(1)} \tag{16}$$

**Final Gradient with Respect to Input Variables** Combining all the terms from the chain rule equations 5, 13, 14, 15, 16 the gradient of the loss function $L$ with respect to the input variable $x_d$ is:

$$\frac{\partial L}{\partial x_d} = \sum_{l=1}^{L} \sum_{i=1}^{M_l} \frac{2}{m} (\hat{y}_i - y_i) \sum_{n=1}^{N_l} w_{l,i,n} \frac{\partial B_{N_l,p_l,n}(z_i^{(l)})}{\partial z_i^{(l)}} W_{id}^{(1)} \tag{17}$$

The gradient of the pre-activation with respect to the weight is:

$$\frac{\partial z_i^{(l)}}{\partial W_{ij}^{(l)}} = h_j^{(l-1)} \tag{18}$$

### A.3 FLOPs and Parameter Comparison

This appendix analyzes the performance of MLP, KAN, and LCN models across various datasets in terms of FLOPs (floating-point operations per second) and accuracy. The comparisons highlight the computational efficiency and effectiveness of these architectures.

#### A.3.1 Visualization of FLOPs and Accuracy

Figures 6 and 7 present the accuracy versus FLOPs for six datasets: Bank, Bean, Spam, Telescope, MNIST, and FMNIST. These figures illustrate the trade-off between computational cost and performance for MLP, KAN, and LCN models.

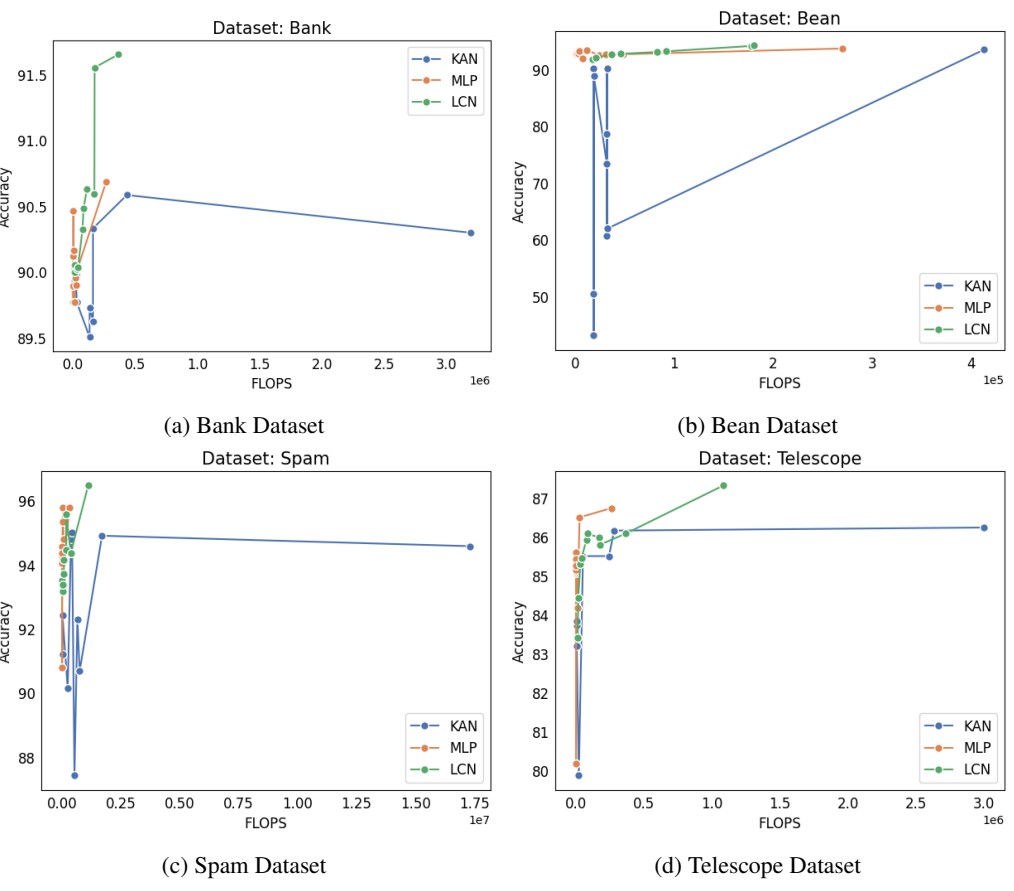

(a) Bank Dataset

(b) Bean Dataset

(c) Spam Dataset

(d) Telescope Dataset

Figure 6: Comparison of accuracy versus FLOPs for MLP, KAN, and LCN models on Bank, Bean, Spam, and Telescope datasets.

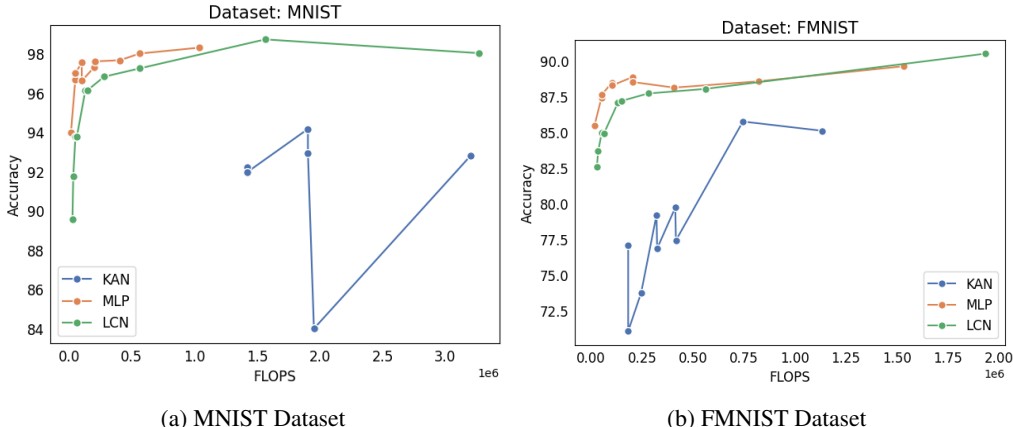

(a) MNIST Dataset                    (b) FMNIST Dataset

Figure 7: Comparison of accuracy versus FLOPs for MLP, KAN, and LCN models on MNIST and FMNIST datasets.

### A.3.2  INSIGHTS AND OBSERVATIONS

**Bank and Bean Datasets**   For the Bank and Bean datasets, LCN consistently achieves higher accuracy with fewer FLOPs compared to KAN. While MLP demonstrates efficiency in FLOPs, it underperforms in accuracy, particularly on the Bean dataset. LCN's localized activation functions enable a better trade-off between computational efficiency and accuracy.

**Spam and Telescope Datasets**   On the Spam and Telescope datasets, LCN surpasses both KAN and MLP in accuracy. KAN's higher FLOPs are a result of its complex architecture, while LCN leverages its B-spline-based activations to achieve efficient learning with reduced computational resources.

**MNIST and FMNIST Datasets**   For MNIST and FMNIST, LCN demonstrates scalability and adaptability to high-dimensional inputs, maintaining an advantage in accuracy with comparable or lower FLOPs than KAN. MLP struggles to achieve the same level of performance, further emphasizing the effectiveness of LCN's design.

### A.3.3  CONCLUSION FROM FLOPS ANALYSIS

The FLOPs analysis highlights LCN's ability to balance computational cost and accuracy effectively. Across all datasets, LCN consistently outperforms KAN in efficiency and accuracy while achieving higher accuracy than MLP. This makes LCN a robust and scalable model for diverse machine learning tasks.

