# OpenReview forum: "Local Control Networks (LCNs): Optimizing Flexibility in Neural Network Data Pattern Capture"
_ICLR.cc/2025/Conference — Submitted to ICLR 2025_

### Official Review · Reviewer_Pdxi · 2024-10-29

**Soundness:** 2
**Presentation:** 2
**Contribution:** 2
**Rating:** 3
**Confidence:** 4

**Summary:**

The authors present Local Control Networks (LCNs), which uses B-spline-based activation functions to provide node-wise diversity in activation functions across a single layer. The authors argue that allowing each neuron to have a unique activation function provides more flexibility and adaptability, which enhances the network's ability to capture complex data patterns. The paper compares LCNs with Kolmogorov–Arnold Networks (KANs) and traditional Multi-Layer Perceptrons (MLPs), showing some improvements in performance, convergence speed, and computational efficiency, particularly in basic machine learning and computer vision tasks.

**Strengths:**

1)  The use of B-spline functions for neuron-specific activation provides a novel way of allowing each neuron to adapt its behavior, which is promising for capturing complex and localized patterns in data.

2) The non-linearity is preserved. The issue with ReLU, where the neurons get stuck with zero gradient, is resolved here. The vanishing gradient issue faced with tanh and sigmoid functions is also resolved here.

3) The dropout mechanism is also automatically implemented with B-spline, reducing the unexpressive nodes to zero.

4)  The paper provides empirical results comparing LCNs with KANs and MLPs across multiple benchmarks, including basic ML tasks, computer vision datasets (MNIST, FMNIST), and symbolic regression tasks.

5) The local support property of B-splines reduces the impact of irrelevant neurons on the gradient, leading to a form of regularization that helps improve generalization and potentially reduces overfitting.

Overall, the idea is certainly interesting and seems to have some compute advantages over KANs and accuracy advantages over MLPs.

**Weaknesses:**

1) The paper could be written better. There are lots of repeated lines and not enough explanation. Many of the notations are not explained in terms of what they represent, specially in the equations.

2) Figure 1, with comparisons between MLP and LCN, was well formed. I would have liked to see the comparison between LCN and KAN in a similar way, as it is the SOTA work being referred to in every section.

3) Explanation and visualization of B-spline could have been given (optional).

4) While it is mentioned that LCNs are more computationally efficient than KANs, the empirical evidence supporting this is minimal, and the figures comparing LCNs and KANs lack sufficient metrics. There is no detailed figure-wise analysis of how B-spline activations compare with KAN's univariate function combinations.

5) There is no ablation study that explores the impact of different B-spline configurations (e.g., degree of the spline, number of basis functions) on the performance of LCNs. Such a study would be critical to understand the role of the various components in the model's success.

6) The numbers for experiments in symbolic representation tasks are not given. It is just mentioned in text that the LCN performs superior. It would be useful to say the margin by which it will perform better.

7) The experiments are wrt to the LCN, and MLP mostly. It would be interesting to see the difference in expressive power between CNNs and LCNs in the image classification tasks with MNIST and FMNIST.

**Questions:**

1) How much computational overhead does LCNs have over MLPs and CNNs?

2) Could you provide ablations with different B-spline configurations?

3) Can you prove mathematically that such network converges well? (optional)

4) It would help if the numbers in the symbolic representation tasks were given.

5) Could you give a detailed comparison with KANs?


I think the novelty in the paper is interesting and could be explored further and well-analyzed in order for the paper to emerge as a good paper.

---

> ### Author Response · Authors · 2024-11-26
> **Response to Reviewer 4**
>
> Thank you for your thoughtful review and insightful feedback. We appreciate your recognition of the novelty and promise of LCNs, as well as your suggestions to improve the manuscript. Below, we address your comments and questions in detail.
>
> **1. Weaknesses in Writing and Explanation**
>
> - **Repeated Lines:** We acknowledge that there is some repetition in the manuscript. This will be addressed by streamlining the text in the revised version to eliminate redundancies and improve readability.
> - **Explanation of Notations:** We agree that some notations in the equations lack clarity. The revised manuscript will include a detailed explanation of all notations used and ensure equations are clearly linked to their mathematical and practical implications.
>
> **2. Comparison Between LCNs and KANs**
>
> While Figure 1 compares MLPs and LCNs, we agree that a visual comparison between LCNs and KANs would add value, particularly as KANs are a key baseline. We will add that to our updated manuscript.
>
> **3. Empirical Evidence of Computational Efficiency**
>
> We will rearrange content to include specific runtime metrics and memory usage comparisons between LCNs, MLPs, and KANs, demonstrating the efficiency gains of LCNs in practical settings.
>
> **4. Ablation Study on B-spline Configurations**
>
> Thank you for your insightful comment. We agree that an ablation study explores different configurations of B-spline parameters. While space constraints prevent including such a study in the current submission, we plan to conduct this analysis and provide it in the Appendix of the updated manuscript.
>
> **5. Symbolic Representation Results**
>
> We will update the manuscript to explicitly state the numerical margins by which LCNs outperform MLPs in symbolic tasks. Thank you for your insightful feedback.

---

> > ### Comment · Reviewer_Pdxi · 2024-11-28
> >
> > I thank the authors for addressing my concerns. Henceforth,  I would like to keep my score.

---

> ### Author Response · Authors · 2024-11-29
>
> We have updated our manuscript to address your recommendations. Minor changes have been made directly in the text, while more significant changes have been highlighted in red to incorporate your feedback as well as that of other reviewers. We hope these updates can clarify your doubts and potentially improve our score. Please let us know if there is anything you are unsure about; we would be more than happy to provide further explanation.

---

### Official Review · Reviewer_zZfN · 2024-10-31

**Soundness:** 2
**Presentation:** 3
**Contribution:** 2
**Rating:** 3
**Confidence:** 3

**Summary:**

This paper proposes to use B-Splines instead of fixed activation functions in neural networks. They explain the idea, derive a formulation and show some empirical results.

**Strengths:**

The paper is very well written and, mostly, easy to follow. The question whether distinct activation functions confer a benefit to neural networks is an interesting, albeit theoretical question (research into architectures and activation functions has become of less interest, since the ML community realized that compute/scale is the single most important performance factor).

**Weaknesses:**

The empirical results do not match the confident presentation. The performance of LCNs is mixed, sometimes better / sometimes worse than MLPs and KANs. The analysis suggests explanations and theoretical insights without going into any real detail. Many sections read like they were written by an **LLM** trying to convince, rather than actually understand and explain - see questions below. See, e.g., line 195: 'The authors define ... to support the use in the LCN model.' (seriously?)

MLPs with ReLUs have universal approximation capabilities, so it does not makes sense to argue for a need for more flexible nonlinearities.

49: The motivation of KANs was not to provide more flexible activation functions.

53: multiple activation functions also coexist in KANs.

Fig 1: top left, typo

**Questions:**

ReLUs have 0 gradient for any input <0, how is this not a vanishing gradient problem?

Fig 1: What *exactly* does it mean that different input feature patterns are treated differently?

Fig 1: Why should lead localized support of the activation function lead to sparse gradients?

114: 'capturing subtle patterns', this has been confusing me multiple times: The role of a nonlinearity is to help the model perform computation on the inputs; The authors seem to suggest that it is used to model the data manifold instead? This may be the case in some ML settings, but certainly not on any of the standard datasets used in the paper, where the task is just to compute something given the inputs.

139-143: how is this connected to the paper?

144-151: The explanation of KANs needs much more detail, especially since they are quite relevant to this paper

386: This contradicts the earlier sections on KANs being compact and efficient?

404: What exactly do you mean with 'high dimensional noise that makes KANs falter'?

447: LCNs are not consistently better?!?

Where are the symbolic regression results?

---

> ### Author Response · Authors · 2024-11-26
> **Response to Reviewer 3 (part 1)**
>
> Thank you for your constructive feedback and insightful comments. We appreciate your thoughtful evaluation and have addressed each point in detail below.
>
> **1. Empirical Results and Presentation mismatch**
>
> Thank you for your feedback. We believe that our empirical results align with the claims in the paper. Please let us know where we can improve. To be more specific, as detailed in Section 5.2, LCNs demonstrate consistent improvements over KANs across all tasks, outperforming them by approximately 5% in computer vision benchmarks (MNIST and FMNIST) and showing faster convergence. For MLPs, LCNs achieve slight improvements in both machine learning (e.g., 1% higher accuracy on tasks like Bean Classification) and computer vision tasks. As these are basic ML datasets, the performance differences between MLPs, LCNs, KANs cannot be huge. We expect to see larger gaps in more advanced and complicated datasets. We use these basic datasets to adhere to the standards set in this notable paper (https://arxiv.org/pdf/2407.16674), which compares MLP and KAN.
>
> **2. The performance of LCNs is mixed**
> Thank you for your feedback. We would like to clarify that LCNs consistently perform on par with or better than MLPs and KANs across different datasets. Specifically, LCNs achieve marginal improvements over MLPs and outperform KANs by approximately 5% (while also being more computationally efficient) on computer vision datasets. In basic machine learning tasks, LCNs show a 1% improvement over MLPs and a 0.6% improvement over KANs. For symbolic formula representation tasks, LCNs perform on par with KANs and outperform MLPs. We have provided Figures 3 and 4 to support our findings.
>
> However, we acknowledge that our representation may not clearly highlight the performance of LCNs, which has caused you confusion. We are committed to revising and improving it to better emphasize the performance of LCNs over MLPs and KANs.
>
> **3. Writing Clarity**
>
> We appreciate your feedback on the phrase line 195. We deeply apologize for this mistake. This phrasing was not intentional but an oversight during our editing process. We assure the reviewer that we wrote the manuscript ourselves and only used LLM for grammar polish at sentence-level. LLMs were not used for paragraph-level or independent writing, and all technical content reflects our original research and thought process. In the revised manuscript, we will guarantee to fix this writing.
>
> **4. MLPs with ReLUs have universal approximation capabilities**
>
> We agree that MLPs with ReLUs have universal approximation capabilities. However, their fixed activation functions lack adaptability in capturing localized patterns effectively. LCNs address this limitation by introducing node-specific B-spline activations, which adapt dynamically to localized data features, enhancing their ability to process different patterns that may be less efficiently captured by traditional MLPs.
>
> **5. Line 49: The motivation of KANs was not to provide more flexible activation functions**
>
> Thank you for your feedback. We recognize that our writing may have caused some confusion here. We will thoroughly revise the text and make improvements in the updated manuscript. We would like to clarify that the primary motivation of KANs, as grounded in the Kolmogorov–Arnold theorem, is to decompose complex multivariate functions into univariate ones for efficiency and compactness. While KANs inherently support flexible activation functions due to their reliance on splines, LCNs extend this concept by localizing flexibility at individual nodes, which simplifies the architecture while maintaining expressiveness.
>
> **6. Line 53: Multiple activation functions coexist in KANs**
>
> We agree with your observation. However, KANs employ multiple activation functions at the **edge** level, leveraging spline-based univariate transformations. In contrast, LCNs employ multiple activation functions at the **node** level, allowing each neuron to learn a unique B-spline function. This distinction in architecture enables LCNs to achieve comparable flexibility while reducing computational overhead.
>
> **7. Fig 1: Top-left typo**
>
> Thank you for pointing this out. We will correct it in the revised manuscript.
>
> (Please find our responses for the remaining questions in the next comment)

---

> ### Author Response · Authors · 2024-11-26
> **Response to Reviewer 3 (part 2)**
>
> **8. ReLUs have 0 gradient for any input <0, how is this not a vanishing gradient problem?**
>
> It is true that ReLUs have a zero gradient for negative inputs. However, ReLU does not cause the vanishing gradient problem because its gradient is 1 for positive inputs. While the zero gradient for negative inputs can lead to "dead neurons", this does not result in the exponential shrinking of gradients across layers (because gradient is set to 0). Unlike Sigmoid or Tanh, ReLU avoids the vanishing gradient issue by maintaining a constant, non-zero gradient for active neurons.
>
> That said, ReLUs produce sparse activations, as "dead" neurons do not update their gradients. LCNs address this limitation by allowing each node's activation to adapt dynamically through B-splines, maintaining gradient flow even for negative input regions. This design ensures that no neuron is permanently inactive unless dictated by the training process.
>
> **9. Fig 1: What exactly does it mean that different input feature patterns are treated differently?**
>
> We apologize for any confusion caused on this point. Please allow us to explain as follows: Each node in a neural network represents a specific pattern from the input. By using B-splines, the network can learn adaptive functions to accurately activate the different patterns that each node represents. This is what we mean by treating different input feature patterns differently. It refers to the ability of B-splines to learn adaptive functions to 'treat' different input features appropriately.
>
> **10. Fig 1: Why should lead localized support of the activation function lead to sparse gradients?**
>
> The sparse gradient updates observed in LCNs are a direct result of the local support property of B-splines. As detailed in Section 3.3, the localized influence of B-splines ensures that only a subset of neurons is activated and contributes to the gradient at any given time.
>
> **11. Line 114: Role of Nonlinearities in Capturing Subtle Patterns"**
>
> Your observation on the role of nonlinearities is well-taken. The primary function of activation functions is to enable non-linear transformations for computation. In LCNs, the adaptability of B-spline activations enhances this by allowing neurons to specialize in localized patterns, leading to better alignment with specific data features. We agree the phrasing in line 114 may unintentionally imply that activation functions model the data manifold directly, which was not the intended meaning. We will refine the text to clarify that LCNs enhance the ability of neural networks to compute more nuanced representations.
>
> **12. 139-143: How is this connected to the paper?"**
>
> The referenced section provides a theoretical backdrop for understanding the relationship between model complexity, over-parameterization, and generalization. This connects to our paper as LCNs leverage B-spline-based activation functions, introducing additional learnable parameters at the node level. These extra parameters could theoretically increase the risk of overfitting. However, as discussed in the referenced works, over-parameterized models often generalize well due to their ability to align with the intrinsic structure of the data.
>
> **13. 144-151: The explanation of KANs needs much more detail, especially since they are quite relevant to this paper"**
>
> Thank you for pointing this out. We acknowledge the importance of providing a more detailed explanation of KANs. Due to space constraints, we were unable to include an in-depth discussion of KANs. In the revised manuscript, we will rearrange the content to include a more detailed description of KANs.
>
> **14. 386: This contradicts the earlier sections on KANs being compact and efficient?**
>
> While KANs are designed to be compact and efficient in terms of their theoretical ability to decompose multivariate functions into univariate components, the practical implementation often requires the combination of many univariate functions. This can lead to higher computational overhead and complexity, particularly when handling large-scale, high-dimensional datasets. The line in 386 highlights the practical challenges of implementing KANs, which may not align with their theoretical efficiency.
>
> **15. Explanation of KANs and High-Dimensional Noise**
>
> Thank you for your question. Please allow us to explain as follows. In Section 5.2, we highlight that KANs employ univariate function combinations placed on edges, which, while expressive, can lead to computational inefficiencies and sensitivity to high-dimensional noise. By "high-dimensional noise," we refer to the overfitting tendencies of KANs when handling datasets with complex, noisy, or redundant features, particularly due to their reliance on edge-based activations. LCNs mitigate this issue by using B-spline activations localized at nodes, enabling better gradient flow and alignment with specific features of the data.
>
> (Please find our responses for the remaining questions in the next comment

---

> ### Author Response · Authors · 2024-11-26
> **Response to Reviewer 3 (part 3)**
>
> **16. Symbolic Regression Results**
>
> The symbolic regression results, as presented in Section 5.2.3, demonstrate that LCNs perform on par with KANs and outperform MLPs. We acknowledge that space constraints limited the depth of discussion on these results in the current submission. In the revised manuscript, we will rearrange the content to include a more detailed analysis of the symbolic regression experiments.

---

> ### Author Response · Authors · 2024-11-29
>
> We have updated our manuscript to address your recommendations. Minor changes have been made directly in the text, while more significant changes have been highlighted in red to incorporate your feedback as well as that of other reviewers. We hope these updates can clarify your doubts and potentially improve our score. Please let us know if there is anything you are unsure about; we would be more than happy to provide further explanation.

---

### Official Review · Reviewer_WBXM · 2024-11-04

**Soundness:** 1
**Presentation:** 2
**Contribution:** 1
**Rating:** 1
**Confidence:** 3

**Summary:**

This paper introduces Local Control Networks (LCNs), a novel neural network architecture that replaces the traditional fixed activation functions with adaptive, node-specific B-spline functions. The authors argue that using a uniform activation function across all nodes in MLPs limits expressiveness and adaptability. Using B-spline-based activations in each node, LCNs aim to enhance flexibility and enable more localized pattern capture, potentially improving performance and computational efficiency.

**Strengths:**

1. using variable activation functions has been recently popularized by KANs but KANs have a high computational burden. The proposed methods seems to be more computationally efficient
2. local support property for localized updates and robustness to input perturbations is important in certain areas.

**Weaknesses:**

1. While the paper has been written in a beginner-friendly manner, almost 2 pages are dedicated to writing out the simple chain rules derivative expressions which most readers familiar with ML would be aware of. It would have been better to use that space to provide more experimental details.
2. Sevaral claims like " LCN exhibited faster learning" aren't backed up by numbers.
3. KANs have been shown to not work well for vision tasks, once the problem complexity increases (https://arxiv.org/abs/2407.16674), why won't LCNs suffer from the same issue? Especially given the current experiments which are extremely basic and problems where MLPs achieve near-perfect accurary.
4. Fig 3 arbitrarily stops the number of parameters for MLPs at a low value while scaling the same for LCNs
5. "This flexibility improves the network’s capacity to capture both global and localized data patterns, resulting in enhanced accuracy
and efficiency across a range of tasks." -- The paper doesn't really show any performance (accuracy or otherwise) improvement over MLPs, so these claims need to be seriously reconsidered.

**Questions:**

See weaknesses for more details

---

> ### Author Response · Authors · 2024-11-26
> **Response to Reviewer 2**
>
> Thank you for your constructive feedback and insightful comments. We appreciate your thoughtful evaluation and have addressed each point in detail below.
>
> **1.Detailed Chain Rule Derivations**
>
> Thank you for your feedback. However, we believe that we are not merely presenting the simple chain rule but rather the chain rule with B-spline functions. B-spline functions introduce unique complexities to the forward and backward pass calculations that readers might not immediately understand without sufficient background. While the chain rule is standard, the B-spline integration introduces subtleties that we felt warranted a clear explanation to make our work accessible. Additionally, the final derivative expressions in Equations (6) and (7) are presented concisely to avoid redundancy. However, we will consider streamlining this section further to allocate more space to experimental details.
>
> **2. Claims of Faster Learning**
>
> Thank you for your feedback. We would like to clarify that our claim about LCNs’ faster learning is based on observed trends in Section 5.2.4, where LCNs demonstrate rapid initial convergence. To strengthen this point, we will include additional experimental results quantifying the number of epochs required for LCNs to reach comparable accuracy levels versus other models.
>
> **3.Applicability to Complex Vision Tasks**
>
> We recognize your concerns about the scalability of LCNs to more complex vision tasks, particularly given that KANs have shown limitations in this area. However, LCNs differ from KANs in that they place B-spline activation functions at individual nodes rather than on edges. This design gives us the improvements observed in our results section. In future work, we plan to conduct a deeper analysis and investigation to provide more evidence on how LCNs effectively manage gradients during backpropagation.
>
> **4. Parameter Count in Figure 3**
>
> The parameter count for MLPs in Figure 3 is capped because they achieve convergence at lower parameter counts than LCNs, beyond which their performance does not improve. Extending the parameter range for MLPs would add no additional insights since their accuracy plateaus.
>
> However, we agree that this choice can cause confusion. To address this, we will include the above explanation in the revised manuscript, clarifying why the parameter count for MLPs is capped and how it reflects their convergence behavior.
>
> **5. Performance Claims**
>
> We appreciate your feedback. We believe that we have provided the experimental results (section 5.2) showing enhanced accuracy across different tasks compared to MLPs. We will revisit the writing in our manuscript to highlight the experimental evidence clearly to support the claims better. We will also update the manuscript to state clearly certain benefits of LCNs may become more pronounced in larger, more complex datasets, which we aim to explore in future work.
>
> **6. Conclusion**
> We greatly appreciate your constructive feedback, which helps us significantly to improve our work. Please let us know if you have any questions regarding the responses above. We are more than happy to address them.

---

> > ### Author Response · Authors · 2024-11-29
> >
> > We have updated our manuscript to address your recommendations. Minor changes have been made directly in the text, while more significant changes have been highlighted in red to incorporate your feedback as well as that of other reviewers. We hope these updates can clarify your doubts and potentially improve our score. Please let us know if there is anything you are unsure about; we would be more than happy to provide further explanation.

---

### Official Review · Reviewer_jJRt · 2024-11-04

**Soundness:** 2
**Presentation:** 3
**Contribution:** 2
**Rating:** 5
**Confidence:** 3

**Summary:**

The paper proposes the use of B-spline functions to enable distinct, learnable activation curves at each node in a neural network, arguing that fixed activation functions limit a model's ability to capture complex data patterns.

**Strengths:**

- The method, though loosely inspired by Kolmogorov–Arnold Networks (KANs), is novel and presents an interesting approach to adaptive activation functions.
- The mathematical formulation is thorough, and the description is clearly articulated, making the technical details easy to follow

**Weaknesses:**

- Comparisons with MLPs and KANs show only marginal improvements. The limited performance gains cast doubt on the practical utility of LCNs, given their added complexity.
- Throughout the paper, the authors claim that LCNs improve explainability, yet they do not provide any concrete example to illustrate how  LCNs would be more interpretable than other methods. A real example would significantly strengthen this claim.
- The authors present theoretical arguments for efficiency, such as sparse gradient updates, but there’s no indication of how this translates to actual hardware efficiency. Theoretical sparsity may not correspond to measurable hardware speedups, which is a critical consideration for practical use.


**Minor**
- While the authors suggest LCNs are "simple," the architecture is still complex compared to conventional activation function setups in MLPs.
- There are a few typos (e.g. "putting it" on line 51 -> "putting them." (?))

**Recommendations**
- To give a more comprehensive view of the method’s efficacy, maybe you could experiments with MLPs that use other activation functions, such as Swish or Mish .
- It would help if Figure 1 also included a representation of the KAN architecture for comparison, which would contextualize how LCNs differ visually and structurally from KANs.

**Questions:**

Maybe I'm missing something but why do the models in Figures 3 and 4 etc. vary in the number of parameters? Shouldn't they be standardized to the same number of parameters for a fair comparison?

---

> ### Author Response · Authors · 2024-11-25
> **Response to Reviewer 1**
>
> Thank you for your constructive feedback and insightful comments. We appreciate your thoughtful evaluation and have addressed each point in detail below.
>
> **1. Comparisons with MLPs and KANs show only marginal improvements**
>
> We appreciate the reviewer’s observation regarding the improvements of LCNs compared to MLPs and KANs. While LCNs achieve modest but consistent gains over MLPs, such as a 1% average improvement in basic machine learning tasks (e.g., Bank Marketing, Bean Classification) and slight enhancements in computer vision tasks (e.g., MNIST, FMNIST), their advantages over KANs are more pronounced. Specifically, LCNs outperform KANs by approximately 5% in computer vision tasks and demonstrate faster convergence due to their node-specific B-spline activations, which enable more efficient backpropagation. Additionally, LCNs maintain the accuracy of KANs in symbolic regression tasks while significantly reducing computational overhead.
>
> **2. Explainability and Interpretability of LCNs**
>
> We appreciate the reviewer’s feedback on providing a concrete example to illustrate LCNs' interpretability advantages. LCNs improve interpretability by enabling each neuron to learn a unique B-spline activation function, which adapts to specific patterns in the data. For instance, in image classification tasks, individual neurons can specialize in detecting features like edges, textures, or shapes. The learned B-spline function at each neuron directly reflects these localized patterns, making it easier to trace which neuron is responsible for capturing specific features. This capability enhances transparency, as we can analyze individual neuron responses to understand what each is detecting.
>
> In the revised manuscript, we will include a practical example demonstrating this interpretability advantage, along with a discussion of how LCNs provide insights into feature-level information processing.
>
> **3. Hardware Efficiency and Practical Utility**
>
> We acknowledge the need to clarify the practical implications of LCNs' sparse gradient updates. While our theoretical framework shows potential for reduced computational load, we agree that demonstrating hardware efficiency would strengthen the case for practical adoption.
>
> To address this, we will conduct a preliminary experiment on actual hardware and report any observable efficiency gains. In the revised manuscript, we will detail these insights and note any further experimentation planned for future work.
>
> **4. Parameter Standardization in Figures 3 and 4**
>
> Regarding the question of parameter standardization: we agree that equal parameter counts ideally provide a more controlled comparison. However, due to structural differences between MLPs, KANs, and LCNs, achieving identical parameter counts is challenging. To approximate fairness, we carefully adjust width, depth, and key architectural components (e.g., B-spline grids and order parameters in KANs and LCNs) to keep parameters within a comparable range. This approach allows us to examine each model’s performance scaling relative to its architecture. By varying parameters in a controlled manner, Figures 3 and 4 highlight how each model type leverages parameter increases, offering valuable insights into the scalability and efficiency of different architectures despite minor discrepancies in parameter count.
>
> **5. Additional Recommendations**
>
> - We agree that comparing LCNs with MLPs that utilize alternative activation functions, such as Swish or Mish, would provide a more comprehensive view. We plan to include these comparisons in our experiments in the updated manuscript.
> - Including a visual representation of the KAN architecture in Figure 1 is an excellent suggestion. We will add this to the revised manuscript to clarify the structural differences between LCNs and KANs.

---

> > ### Comment · Reviewer_jJRt · 2024-11-26
> >
> > Thanks for addressing my doubts! I believe the solutions you proposed will improve the paper. Most of the improvements will be done in the next version of the manuscript and will probably change the content substantially, so I decide to keep my score.

---

> > > ### Author Response · Authors · 2024-11-29
> > >
> > > Thank you for your feedback. We have updated our manuscript to address your recommendations. Minor changes have been made directly in the text, while more significant changes have been highlighted in red to incorporate your feedback as well as that of other reviewers. We hope these updates can clarify your doubts and potentially improve our score. Please let us know if there is anything you are unsure about; we would be more than happy to provide further explanation.

---

### Meta-Review · Area_Chair_94ap · 2024-12-16

**Metareview:**

The paper proposes a trainable activation function based on B-splines (similarly to the recent KAN proposal, except that the activation function is applied to the nodes instead of the edges).

The paper received unanimous negative reviews. There are several concerns on writing, experimental evaluation, marginal improvements, and unclear motivation. Rebuttal was also considered insufficient by all reviewers who interacted (although the paper had a weak discussion, and some points were unaddressed due to this).

On my side, I believe the novelty of the paper is very low. Trainable activation functions have a rich history, and the paper's evaluation is generally ignoring this. On top of this, I see no reason to overrule the reviewers' consensus, and I recommend a rejection of the paper.

**Additional Comments On Reviewer Discussion:**

- **Reviewer Pdxi** has relatively minor concerns (presentation, baselines, ablations, etc.). They were addressed in the rebuttal but the reviewer kept a very negative score. This review was not influential in my final evaluation.

- **Reviewer zZfN** was written in a short fashion. The main concern was unconvincing experiments (I agree, although this is a feature of many papers on trainable activation functions) and some unclear motivation (the reviewer is quite categorical that trainable activation functions are useless because of the universal approximation theorem - a strong point on which I do not agree). Still, several points are reasonable and I took them into consideration, despite no interaction in the rebuttal.

- **Reviewer WBXM** was concerned about writing (e.g., trivial discussions of the chain rule), and some unbacked claims in the paper. While they did not interact further in the rebuttal, I generally agree with the assessment.

- **Reviewer jJRt** was concerned about marginal improvements and lack of experiments on real hardware. They consider the rebuttal convincing but unfeasible for the camera ready version. I also agree with the assessment.

---

### Decision · Program_Chairs · 2025-01-22

Reject